# Assessing elevated pressure impact on photoelectrochemical water splitting via multiphysics modeling

Feng Liang [1], Roel van de Krol [1,2] & Fatwa F. Abdi [1,3] ✉

Photoelectrochemical (PEC) water splitting is a promising approach for sustainable hydrogen production. Previous studies have focused on devices operated at atmospheric pressure, although most applications require hydrogen delivered at elevated pressure. Here, we address this critical gap by investigating the implications of operating PEC water splitting directly at elevated pressure. We evaluate the benefits and penalties associated with elevated pressure operation by developing a multiphysics model that incorporates empirical data and direct experimental observations. Our analysis reveals that the operating pressure influences bubble characteristics, product gas crossover, bubble-induced optical losses, and concentration overpotential, which are crucial for the overall device performance. We identify an optimum pressure range of 6–8 bar for minimizing losses and achieving efficient PEC water splitting. This finding provides valuable insights for the design and practical implementation of PEC water splitting devices, and the approach can be extended to other gas-producing (photo)electrochemical systems. Overall, our study demonstrates the importance of elevated pressure in PEC water splitting, enhancing the efficiency and applicability of green hydrogen generation.

Solar water splitting has been considered a technology that could potentially overcome the limitation of sunlight's intermittency by storing solar energy in the form of hydrogen. Two main approaches exist, an indirect approach in which a photovoltaic cell is coupled to an electrolyzer (PV-EC) and a direct approach in which photoelectrodes are used to combine the functions of light absorption and electrocatalysis in a single photoelectrochemical (PEC) device. The latter offers several advantages. First, the close integration of the light absorber with the electrocatalyst in a PEC cell greatly improves the system's thermal management. For PV devices, the thermal loss from solar radiation can lead to an operating temperature of 60–80 °C, resulting in an efficiency loss of ≥10% within the semiconductors[1]. In PEC devices, however, such detrimental temperature will not be reached because the surrounding electrolyte acts as a coolant for the

electrodes. Moreover, any temperature increment will both reduce the thermodynamically required voltage for water splitting[2–6], and enhance the electrochemical reaction kinetics and mass transport in the electrolyte[7–10]. An increase in temperature may also enhance charge transport within the photoelectrode material itself, particularly for polaronic materials such as metal oxides, leading to higher efficiencies[9,10]. Second, PEC devices show much lower operating current densities (10–20 mA cm$^{-2}$) than commercial water electrolyzers (0.5–2 A cm$^{-2}$). This dramatically reduces the demands on the electrocatalysts and may enable the use of earth-abundant alternatives to platinum-group metals (PGMs), such as Ru, Ir, and Pt[11–13].

With significant progress made in developing efficient light-absorbing semiconductors and electrocatalysts[14–22], reactor or device engineering currently plays a critical role in translating the materials

[1]Institute for Solar Fuels, Helmholtz-Zentrum Berlin für Materialien und Energie GmbH, Hahn-Meitner-Platz 1, Berlin, Germany. [2]Technische Universität Berlin, Department of Chemistry, Straße des 17. Juni 124, Berlin, Germany. [3]School of Energy and Environment, City University of Hong Kong, 83 Tat Chee Avenue, Kowloon, Hong Kong SAR, China. ✉e-mail: ffabdi@cityu.edu.hk

and devices developed in laboratories towards practical applications. An important parameter in the operation of the PEC device is the operating pressure. All PEC water-splitting devices reported in the literature operate at atmospheric pressure. However, operating PEC water-splitting devices at elevated pressure offers several practical benefits. First, most applications that use hydrogen (e.g., hydrogen-powered vehicles, petroleum refining, ammonia production, and methanol synthesis) require the hydrogen to be stored or processed at elevated pressure. Directly generating hydrogen at a higher pressure in PEC cells would remove the need for extra pressurization steps or at least reduce the number of compression stages[23]. Second, operation at elevated pressure is expected to decrease bubble formation. This is beneficial since extensive bubble formation may lead to optical losses due to scattering and diffraction[24,25], ohmic losses in the electrolyte[26–28], as well as severe product crossover in the absence of adequate electrolyte flow[29,30]. Recently, the concept of in situ utilization of photoelectrochemically generated $H_2$ for hydrogenation of biomass feedstock was introduced and demonstrated as an approach to increase the competitiveness of the PEC system[31]. Elevating pressure in such a system would increase the concentration of dissolved $H_2$, thus potentially facilitating the hydrogenation reaction.

At the same time, operating PEC cells at elevated pressure can also have drawbacks. Although decreased bubble formation is generally considered to be favorable, it may diminish the effective role of bubble-induced micro- and macro-convection in ensuring efficient mass transport and stabilizing the local pH near the electrodes[32–34]. Another drawback is the increase in thermodynamic voltage that is needed when splitting water at elevated pressure[35,36]. A 100 to 120 mV increase of electrolyzer cell voltage, attributed to the Nernst potential, has been reported with increasing operating pressure from 1 to 50 bar[35].

In the present study, we investigate the benefits vs. penalties associated with PEC water splitting at elevated pressure. A membrane-free device configuration is considered, in which hydrogen and oxygen separation is ensured via proper hydrodynamic control[37,38]. This separator-free approach has been demonstrated for various electrochemical devices, such as water electrolyzers, fuel cells, and redox flow batteries[30,39–45]. We develop a multiphysics model that accounts for the electrochemistry, two-phase fluid flow, and mass transport in a PEC device, with a particular focus on the pressure-dependent bubble characteristics. Two main parameters, the bubble formation efficiency, $\eta_{bub}$, and the average bubble diameter, $D_{bub}$, are obtained from empirical data and validated by experimental observation of bubbles formed in a pressurized water-splitting cell. Using this model, we quantify the impact of increasing the operating pressure on the various loss mechanisms (e.g., optical scattering, product crossover, concentration overpotential) in a membrane-free PEC water-splitting device. By considering these losses together with the thermodynamic voltage penalty and the specific work needed for hydrogen compression, the benefits vs. penalties are analyzed, and the optimum operating pressure range for membrane-free PEC water-splitting devices is determined.

## Results and Discussion
### Pressure-dependent bubble characteristics
Bubble formation in a (photo)electrochemical system is a complex process, as it is affected by various factors, such as current density[46], electrolyte flow rate[47], surface tension of the electrolyte[48,49], concentration of the electrolyte[50], hydrophobicity of the electrodes[51], and operating pressure[28,52]. Herein, we correlate the bubble characteristics with the operating pressure $p$, current density $\mathbf{j}_{loc}$, and electrolyte flow velocity $\mathbf{v}_L$ through a multivariable regression of the extensive experimental data reported by Sillen[28]. To the best of our knowledge, this is the only study that reported bubble characterizations in an electrochemical device operating at higher pressure. In that study,

both oxygen and hydrogen bubbles generated under wide operating conditions ($\mathbf{j}_{loc} = 50 – 500\ \text{mA cm}^{-2}$, $p = 0.25 – 30\ \text{bar}$, $\mathbf{v}_L = 0 – 100\ \text{cm s}^{-1}$) were investigated. Interested readers are referred to Chapter 3 in Sillen's report for the detailed experimental conditions[28]. We use a mathematical model of dynamic objects for nonlinear regression based on nonlinear least squares (NLS) to describe the correlation between the target variables and the above-mentioned parameters. A versatile software package (ndCurveMaster) was adopted to conduct the nonlinear regression and find the optimized fitting equations for the average diameter ($D_{O_2}$, $D_{H_2}$) and number density ($N_{O_2}$, $N_{H_2}$) of the oxygen and hydrogen bubbles. The following generic form of the fitting equations was obtained with the respective coefficients listed in Table S1:

$$\left(D_{O_2}, D_{H_2}, N_{O_2}, N_{H_2}\right) = a + b_1 * \mathbf{j}_{loc}{}^{b_2} + c_1 * \mathbf{v}_L{}^{c_2} + d_1 * p^{d_2} + b_3 * \exp\left(\mathbf{j}_{loc}\right)^{b_4}$$
$$+ c_3 * c_4{}^{v_L} + b_5 * \mathbf{j}_{loc}{}^{b_6} + d_3 * \exp\left(p\right)^{d_4}$$

$$(1)$$

Figure 1 compares the calculated values from the fitting equations with the experimental dataset reported by Sillen (black data points). They agree relatively well, as shown by the coefficient of determination ($R^2$, which describes the goodness of fit) of the multivariable regression. It is important to note that the current densities applied in Sillen's experiment are in the range of $50 – 500\ \text{mA cm}^{-2}$, which is considerably higher than the typical current density range in PEC cells ($1 – 20\ \text{mA cm}^{-2}$). We therefore also checked our fitting equations against another dataset reported by Holmes-Gentle et al.[24], in which $O_2$ bubble characteristics were measured at current densities of $0.5 – 8\ \text{mA cm}^{-2}$. These data are added as red data points in Fig. 1a, b, and they remain close to the $y = x$ line; this confirms the applicability of the fitting equations at lower current densities.

As a further confirmation step, we also performed experiments to monitor the generation of $O_2$ bubbles in a moderately pressurized water-splitting cell and determine the bubble formation efficiency ($\eta_{bub}$). We refer to the Methods section for the description and Fig. S2 for the schematic of the experimental setup and the shadowgraphs of the observed bubbles. Figure 2a compares the $\eta_{bub}$ vs. pressure curve for $O_2$ bubbles as determined from our experiments (see Supplementary Note 1 and Fig. S2) to that calculated using Eq. 1. The experimental data fit relatively well with the calculated values, which supports the validity of our derived pressure-dependence relationship of bubble characteristics.

Figure 2b–d shows the $O_2$ and $H_2$ bubble characteristics at various operating pressures, as obtained from our fitting equations. The average bubble diameters ($D_{bub}$) of $O_2$ and $H_2$ significantly decrease with increasing pressure, while the number densities of bubbles ($N_{bub}$) for both gasses increase with pressure elevation. Finally, the derived bubble formation efficiency, $\eta_{bub}$, decreases as pressure increases. This is not surprising since the solubility of $H_2$ and $O_2$ increases at higher pressure, resulting in higher fractions of dissolved $H_2$ and $O_2$ in the electrolyte.

We briefly note that the obtained $\eta_{bub}$ for $H_2$ is considerably lower than that for $O_2$. This is surprising since oxygen is more soluble in water than hydrogen at the same temperature[53,54]. We tentatively attribute this deviation to a higher density of bubble nucleation sites for $H_2$ than $O_2$. Indeed, $H_2$ bubbles are smaller than $O_2$ bubbles, and the density of $H_2$ bubbles is much higher than that of $O_2$ bubbles (see Fig. 2b, c). As a result, the interface area between $H_2$ bubbles and the electrolyte is large, and compared to $O_2$, more $H_2$ molecules can exchange with the (non-saturated) electrolyte and end up as dissolved gas. In the following sections, we use the bubble formation efficiency of $O_2$ for both gaseous products in our simulation. It should be noted that a higher value of the $H_2$ bubble formation efficiency in the model

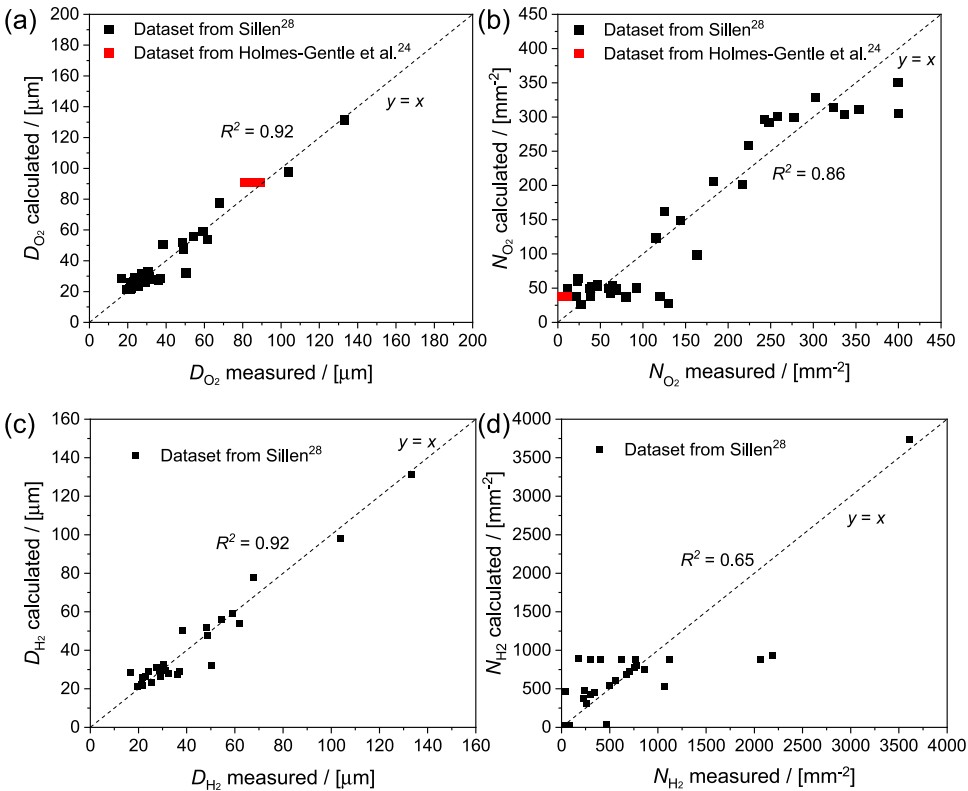

**Fig. 1 | Comparison of bubble properties obtained from calculation and measured values in the literature. a** $O_2$ bubble diameter ($D_{O_2}$) and (**b**) $O_2$ number density ($N_{O_2}$) as calculated from Eq. 1 vs. measured values in the literature. The same plots for $H_2$ bubble diameter and number density are shown in (**c**) and (**d**). The

goodness of the fits is evidenced from the $R^2$ values. We attribute the relatively low $R^2$ value for $H_2$ bubble densities to the rather dispersed datasets; Note that the data for oxygen is distributed across two orders of magnitude, while the hydrogen data spans three orders of magnitude.

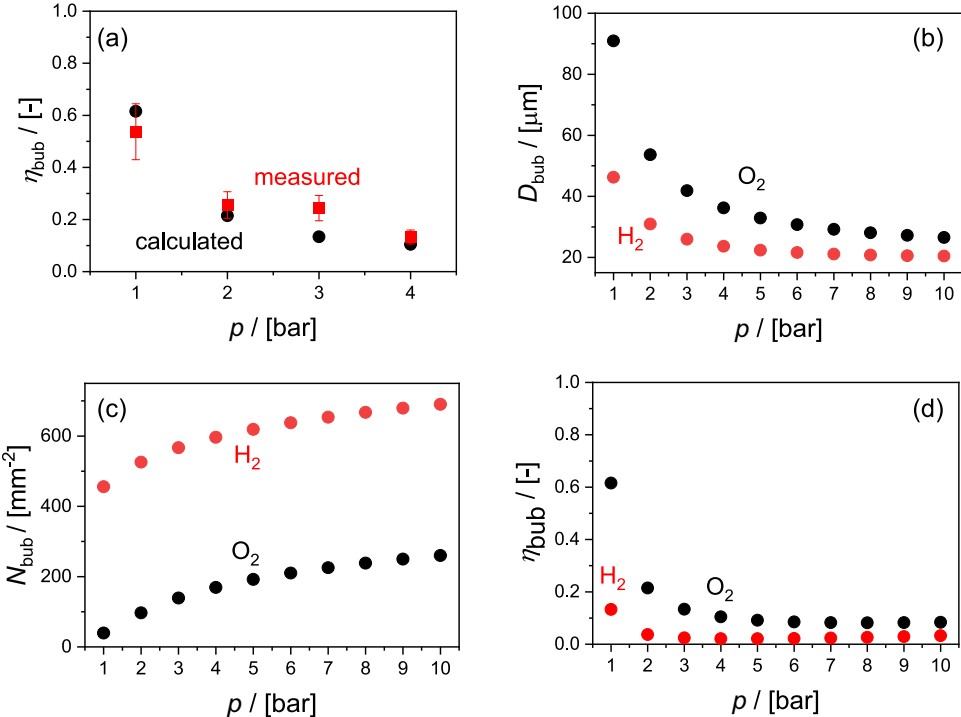

**Fig. 2 | Pressure-dependent bubble characteristics. a** $O_2$ bubble formation efficiency ($\eta_{bub}$) vs. operating pressure. Comparison of the $O_2$ bubble formation efficiency as determined from our experiments and calculated using Eq. 1. The error bars represent ±20% of the measured values, while the values are the average of 500 image frames. (**b**–**d**) Calculated pressure-dependent $O_2$ and $H_2$ bubble

characteristics. **b** Average bubble diameters ($D_{bub}$), (**c**) number density of bubbles ($N_{bub}$), and (**d**) derived bubble formation efficiency ($\eta_{bub}$) as a function of the operating pressure. The current density and the electrolyte velocity are kept constant at 10 mA cm⁻² and 3 cm s⁻¹, respectively.

would lead to an overestimation of the influence of $H_2$ bubbles on e.g., optical losses.

## Influence of pressure on the bubble plume, molar fluxes of products, and crossover

We now investigate the influence of operating pressure on the bubble plume and product crossover in a membrane-free PEC water-splitting device. Schematic representations of our model and the accompanying boundary conditions are shown in Fig. S3. Further details of the modeling are described in the Methods section. In brief, two-dimensional (2D) multiphysics simulations were performed with a specific electrode length (same as channel length $L_y$) and the gap between electrodes (channel width $L_x$, see Table S2). In this 2D model, the directions perpendicular and parallel to the channel flow are defined as the $x$ and $y$ directions, respectively. The device tilt angle, $\theta$, is defined as the angle between the $y$ direction and the horizontal orientation (see Fig. S3a). An Euler-Euler multiphase model was used, which calculates the volume fraction ($\alpha_i$) and velocity vectors ($\mathbf{v}_i$) of phase $i$ (i.e., gas or liquid). The pressure-dependent bubble formation efficiency, $\eta_{bub}$, was applied at the electrode surface as a boundary condition to determine the flux and fraction of the gaseous products (see Figs. S3b, c and Eqs. 10 and 11 in the Methods section). The amounts of dissolved gasses in the electrolyte were therefore calculated from $(1 - \eta_{bub})$ and then further modeled with the transport of diluted species theory. Unless specified differently, the current density was 10 mA cm$^{-2}$, and the average inlet velocity of the electrolyte was 3 cm s$^{-1}$. The pressure dependence of the gas density was included via the ideal gas law in our simulations. An increase in pressure was found to have negligible impact on other gas properties, such as dynamic

viscosity $\mu_g$, specific heat capacity $Cp_g$, and thermal conductivity $k_g$; nevertheless, these properties were also updated with their pressure-dependent values during the simulations[55]. Detailed parameters and properties used in our numerical simulations are tabulated in Tables S2–S4. To ensure the validity of our multiphase flow model, measurements using a flow cell setup were conducted. The description of this validation experiment and the comparison of simulation vs. experimental results are shown in Supplementary Note 2, Fig. S4 and S5, and Supplementary Movie 1–4.

We consider a device configuration in which the anode is the upward-facing electrode (see Fig. S3a for the definition of upward- and downward-facing electrodes). Bubbles produced at the downward-facing electrode (i.e., $H_2$ at the cathode in this case) remain very close to the electrode due to the upward buoyancy force; they are unlikely to contribute to product crossover and are therefore not considered in this section. Detailed discussions on this as well as the choice of anode as the upward-facing electrode for product crossover analysis, have been provided in our previous study[29]. Fig. 3 shows the simulated volume fraction of $O_2$ bubbles in the PEC device at 1 bar and at elevated pressures for the device tilt angle of 90° (a–e) and 30° (f–j). At 1 bar (Fig. 3a, f), the impact of the device tilt angle on the extent of the bubble plume can be clearly observed. When the PEC device is oriented vertically ($\theta$ = 90°), the bubble plume remains very close to the electrode surface (see magnification of this region in Fig. 3a). At $\theta$ = 30° (Fig. 3f), the bubble plume is extended towards and even beyond the middle of the channel. The bubble plume is much suppressed with increasing pressure. At 2 bar, as shown in Fig. 3b, g, the bubble plume remains relatively close to the electrode surface even when the PEC device is tilted at 30°(which is generally considered the

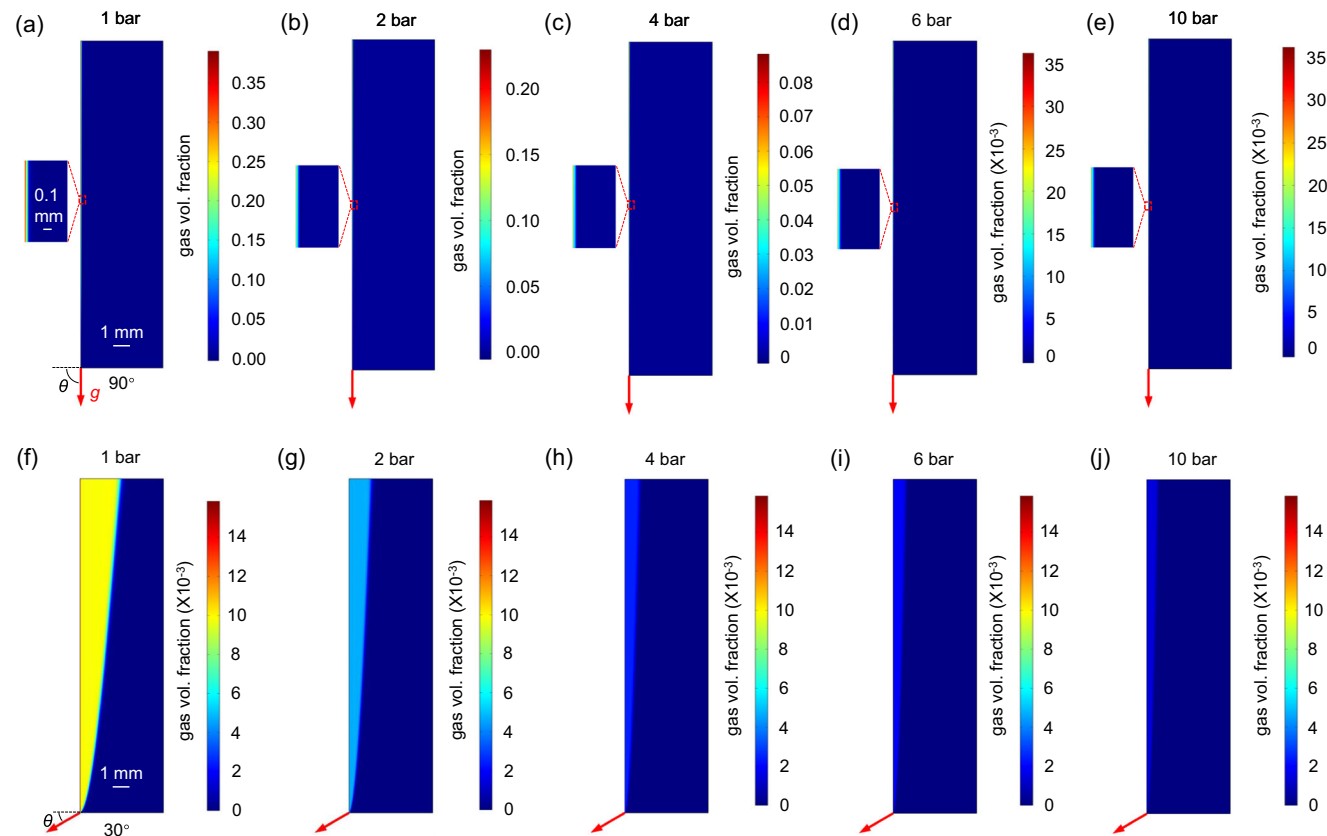

**Fig. 3 | Bubble plume distribution at different operating pressures and device tilt angles.** Colormaps of the simulated gas volume fraction ($O_2$) generated from the anode (left boundary of the domain) in a device tilted at $\theta$ = 90° (**a**–**e**) and 30° (**f**–**j**) and operated at various pressures. (**a**) and (**f**) are at 1 bar, (**b**) and (**g**) are at

2 bar, (**c**) and (**h**) are at 4 bar, (**d**) and (**i**) are at 6 bar, (**e**) and (**j**) are at 10 bar. The current density is 10 mA cm$^{-2}$, and the average inlet velocity of the electrolyte is set as 3 cm s$^{-1}$.

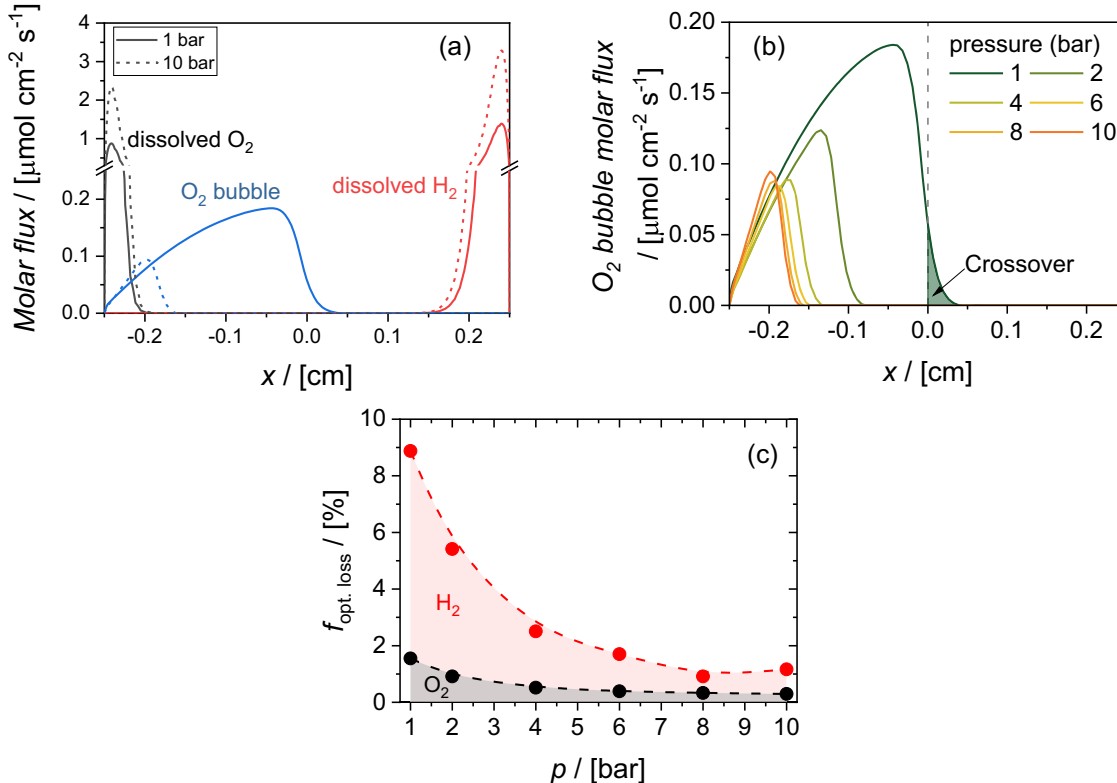

**Fig. 4 | Multiphysics simulation results. a** Simulated molar flux of dissolved products and $O_2$ bubbles at the device outlet under operating pressures of 1 bar (solid) and 10 bar (dashed). **b** Simulated $O_2$ bubble molar flux at the device outlet under various operating pressures. **c** Simulated bubble-plume-induced optical loss ($f_{opt. loss}$) as a function of the operating pressure. The average velocity at electrolyte inlet is set as 3 cm s⁻¹, current density is 10 mA cm⁻², device tilt angle is 30° in (a) and (b), while $\theta = 90°$ in (c).

optimal angle for photovoltaic panels in continental Europe[54]). Further increasing the operating pressure significantly decreases the volume fraction of the gaseous phase, as evident from at least one order of magnitude difference between the color scalebars in Fig. 3b and Fig. 3e. Such a significant effect of pressure is consistent with the lower bubble formation efficiency, $\eta_{bub}$.

To determine the contribution of dissolved and gaseous products to the product crossover, we calculate the molar flux of $O_2$ bubbles and dissolved products at the device outlet, as shown in Fig. 4a, at a device tilt angle of 30°. At 1 bar, the molar flux of the dissolved products (i.e., $O_2$ and $H_2$) remains within -0.1 cm of the electrodes. Increasing the operating pressure, even to 10 bar, does not seem to influence the extent of the molar flux of the dissolved products at the outlet. This is because the molar flux of the dissolved products is dominated by the convective contribution (see Fig. S7); the diffusive contribution is negligible due to the relatively slow diffusion of dissolved $O_2$ and $H_2$ in aqueous solutions[56]. On the other hand, Fig. 4a shows that the molar $O_2$ flux contained in bubbles extends further into the channel and is therefore expected to dominate product crossover. The figure also shows that increasing the pressure from 1 to 10 bar confines the $O_2$ bubbles to a region much closer to the electrode and will thus reduce crossover.

Figure 4b shows the molar flux of $O_2$ bubbles at various pressures. For our membrane-free PEC device configuration, we define product crossover as the fraction of the molar flux that extends beyond the center of the channel (see the shaded region in Fig. 4b), as also similarly considered in other reports[29,39,40,57]. At 1 bar and $\theta = 30°$, the $O_2$ crossover is -1%. As the operating pressure of the PEC device increases, the molar flux of $O_2$ bubbles at the channel outlet is significantly suppressed. For $\theta = 30°$, product crossover is fully prevented for operating pressures higher than 2 bar.

It should be noted that our analysis assumes a uniform bubble diameter in the channel and a constant bubble formation efficiency over the whole electrode surface. In reality, these values are likely to vary across the electrode and channel. Further studies, therefore, require a detailed understanding of bubble nucleation, growth, and detachment to simulate the local-reaction-dependent bubble diameter and bubble formation efficiency more accurately.

## Influence of pressure on the bubble-induced optical loss

As depicted in the schematic representation of the PEC water splitting device (see Fig. S3a), solar photons need to pass the electrolyte before reaching the upward-facing electrode. As the electrolyte contains gaseous products, i.e., $H_2$ and $O_2$, these photons might be scattered and reflected from gas bubbles in the flow, leading to additional optical losses[24,25]. Since performing PEC water splitting at elevated pressure has a significant impact on the bubble formation, we investigate its effect on bubble-induced optical losses.

We quantify the bubble-induced optical losses ($f_{opt.loss}$) based on the volume fraction obtained in our multiphysics simulations. The detailed methods, including the step-by-step procedure and equations, are documented in Supplementary Note 3. We consider both the $H_2$ and $O_2$ bubble plumes in a PEC water-splitting device that is oriented vertically ($\theta = 90°$). This orientation was chosen since Fig. 3 shows that it results in gaseous $H_2$ and $O_2$ volume fractions that are several orders of magnitude higher than other device orientations. A higher volume fraction denotes a denser bubble plume near the electrode, which in turn results in stronger scattering and reflectance of the solar photons. Our analysis, therefore, represents a worst-case scenario for bubble-induced optical losses.

Figure 4c shows the bubble-induced optical losses as a function of operating pressure. At 1 bar, the total optical loss from $H_2$ and $O_2$

bubble plumes is ~9%, and $H_2$ bubbles contribute much more significantly than $O_2$ bubbles. This is because the number density of $H_2$ bubbles, $N_{bub}$, is much higher than that of $O_2$ bubbles (~13 mm$^{-1}$ vs. ~1 mm$^{-1}$ at 1 bar, respectively—see Fig. S10). With increasing pressure, the optical loss decreases significantly, which is consistent with the fact that bubble diameter and bubble formation efficiency both decrease with increasing pressure (see Fig. 2b, d). At 10 bar, the total optical loss is only ~1%, which is almost one order of magnitude lower than the optical loss at 1 bar. This represents an important advantage of operating PEC water-splitting devices at elevated pressure.

We note that a constant scattering coefficient ($K_a = 0.5$) is assumed for both $H_2$ and $O_2$ bubbles in our analysis (see equations S10 and S12). In reality, $K_a$ is affected by the scattering cross section $a_s$ and the bubble cross-section area $a_b$ (see equation S12), which can be associated with bubble characteristics, e.g., bubble shape, size, etc[24,25]. Further correlations between bubbles and the corresponding optical loss can be made using a 'microscopic' point of view that captures time- and spatial-resolved interfacial fluctuations. To do so, more sophisticated numerical models are required, such as Euler-Lagrange[58–60], phase-field[61,62], or Lattice Boltzmann[63–66], but simulations using these models require much higher computational costs and are therefore beyond the scope of the current study.

We briefly acknowledge that the tandem photoelectrochemical water splitting configuration considered in our model, in which two photoelectrodes are used and placed vertically, represents the worst-case scenario for bubble-induced optical scattering loss. In practice, smart engineering of the device configuration, as reported in the literature[23,67], might be applicable to reduce this loss. Another way of alleviating this scattering loss is to employ a "dark" electrode as the upward-facing one and a photoelectrode as the downward-facing one. When this is done, other drawbacks, such as hindered ion transport, the necessity to use noble metals as counter electrodes, etc., need to be carefully considered.

## Influence of pressure on the concentration overpotentials

In our recent study, we showed that bubble-induced convection dominates the fluid dynamics of electrolytes in the vicinity of the electrodes[32]. This is especially important when neutral pH electrolytes are used, in which local pH gradients in electrolyte regions close to the electrode may develop during proton-coupled electron transfer reactions. Since an increase in pressure affects the bubble characteristics, the local pH gradient and the resulting concentration overpotentials are also expected to change with increasing pressure. We, therefore, simulate the impact of increasing pressure on the concentration overpotentials in a PEC water-splitting device with a neutral pH electrolyte (see the "Methods" section for the detailed description of our model). Again, we consider that the PEC water-splitting device is oriented vertically ($\theta = 90°$); a denser bubble plume and a stronger momentum exchange between bubbles and liquid electrolyte are achieved in this orientation, which results in a more pronounced effect on the concentration overpotentials.

Figure 5a shows the colormaps of the pH of the electrolyte (2 M KP$_i$) in a PEC water-splitting device operating at 1 and at 10 bar. The pH gradient at 10 bar is slightly larger than that at 1 bar, although this is difficult to see even in the magnified view of the regions close to the electrodes (Fig. 5a). The pressure-induced effect is more visible shown in Fig. 5b, where the changes in pH ($\Delta$pH) along the surface of the anode and cathode are plotted at various operating pressures. A pH gradient develops along the electrodes, with higher $\Delta$pH observed at regions closer to the outlet ($y = 2$ cm). Increasing the pressure from 1 to 10 bar results in a small increase of the $\Delta$pH. This can be explained by the decreasing bubble diameter and bubble formation efficiency with increasing pressure, which diminishes the beneficial effect of bubble-induced convection in minimizing the pH gradient. However, with only a ~1 mV increase in overpotential after a ten-fold increase in pressure,

the overall effect of pressure on concentration overpotential is negligible (Fig. 5c).

We note that the negligible impact of pressure on the concentration overpotentials may be related to the relatively small size of the cell ($L_x = 0.5$ cm, $L_y = 2$ cm) adopted in the present study. As bubbly flow gradually develops along the electrode (Fig. 5b), the change in concentration overpotential will increase for electrodes that have longer dimensions in the flow direction.

Another important bubble-induced loss is the overpotential due to bubble coverage (or blockage) of the (photo)electrode area. However, this overpotential is not directly solved in our Euler-Euler multi-phase model, as described in the "Methods" section. Instead, we have estimated the bubble coverage overpotential indirectly by (i) integrating the gas volume fraction along the surface of the electrode to calculate the bubble coverage ($\Theta$) and (ii) using the empirical correlation reported by Vogt to evaluate the bubble coverage induced overpotential ($V_\Theta$)[68]. More details on the steps are described in Supplementary Note 4. $\Theta$ and $V_\Theta$ are plotted as a function of pressure in Figs. S11a, b, respectively. Increasing pressure significantly suppresses $\Theta$ and $V_\Theta$; at an operating pressure of 4 bar and higher, the value of $V_\Theta$ is negligible (<5 mV). Note that bubble coverage-induced overpotential is significantly smaller than the pH gradient-induced overpotential. Due to this very minor contribution, $V_\Theta$ is neglected in the following discussions.

## Optimum pressure range for membrane-free photoelectrochemical water splitting

We now combine our findings in the previous sections to determine the optimum pressure range for PEC water-splitting devices. We consider two scenarios in which the end goal is to store $H_2$ at 700 bar (typical pressure of $H_2$ refueling stations), as depicted in Fig. 6a. In the first scenario, PEC water splitting occurs at atmospheric pressure. The generated $H_2$ needs to go through several stages of compression to reach 700 bar. For simplicity, only two stages are illustrated in Fig. 6a: from 1 bar to $p$ and from $p$ to 700 bar. Although more compression stages would realistically be needed, the number of compression stages chosen in our evaluation does not affect the results since we only consider the specific work associated with pressurization. In the second scenario, PEC water splitting is performed directly at an elevated pressure $p$. The need for post-generation compression is therefore reduced, as the generated $H_2$ only needs to be pressurized from $p$ to 700 bar. This advantage, combined with reduced product crossover and reduced bubble-induced optical losses, represents the benefits of the second scenario. On the other hand, directly performing PEC water splitting at elevated pressure in the second scenario results in several penalties, such as higher concentration overpotential and a higher thermodynamic cell voltage (a voltage increase of 48 mV per decade of pressure for both hydrogen and oxygen was reported at 50 °C)[36]. An optimum pressure likely exists at which the benefits of performing PEC water splitting at elevated pressure outweigh the penalties.

In order to make a parallel comparison between the various loss mechanisms, all losses are translated to energy losses in units of kWh. For this purpose, we consider a membrane-free PEC water splitting device with an area of 1 m$^2$ generating 1 kg of $H_2$. The device is oriented vertically, with an inlet electrolyte velocity of 3 cm s$^{-1}$ and an average current density of 10 mA cm$^{-2}$. The specific work needed for compressing 1 kg $H_2$ from the collection pressure $p$ to 700 bar ($f_{swc}$, see the results in Fig. S12) is calculated based on thermodynamics. The crossover loss and the optical loss are translated to kWh ($f_{col}$ and $f_{opl}$, respectively) by considering the lower heating value (LHV) of $H_2$ (33.3 kWh kg$^{-1}$). Finally, the concentration overpotentials and pressure-dependent increment of thermodynamic cell voltage are translated into kWh ($f_{cop}$ and $f_{tcv}$) from mV by taking into account the average current density of the device. Detailed equations on the

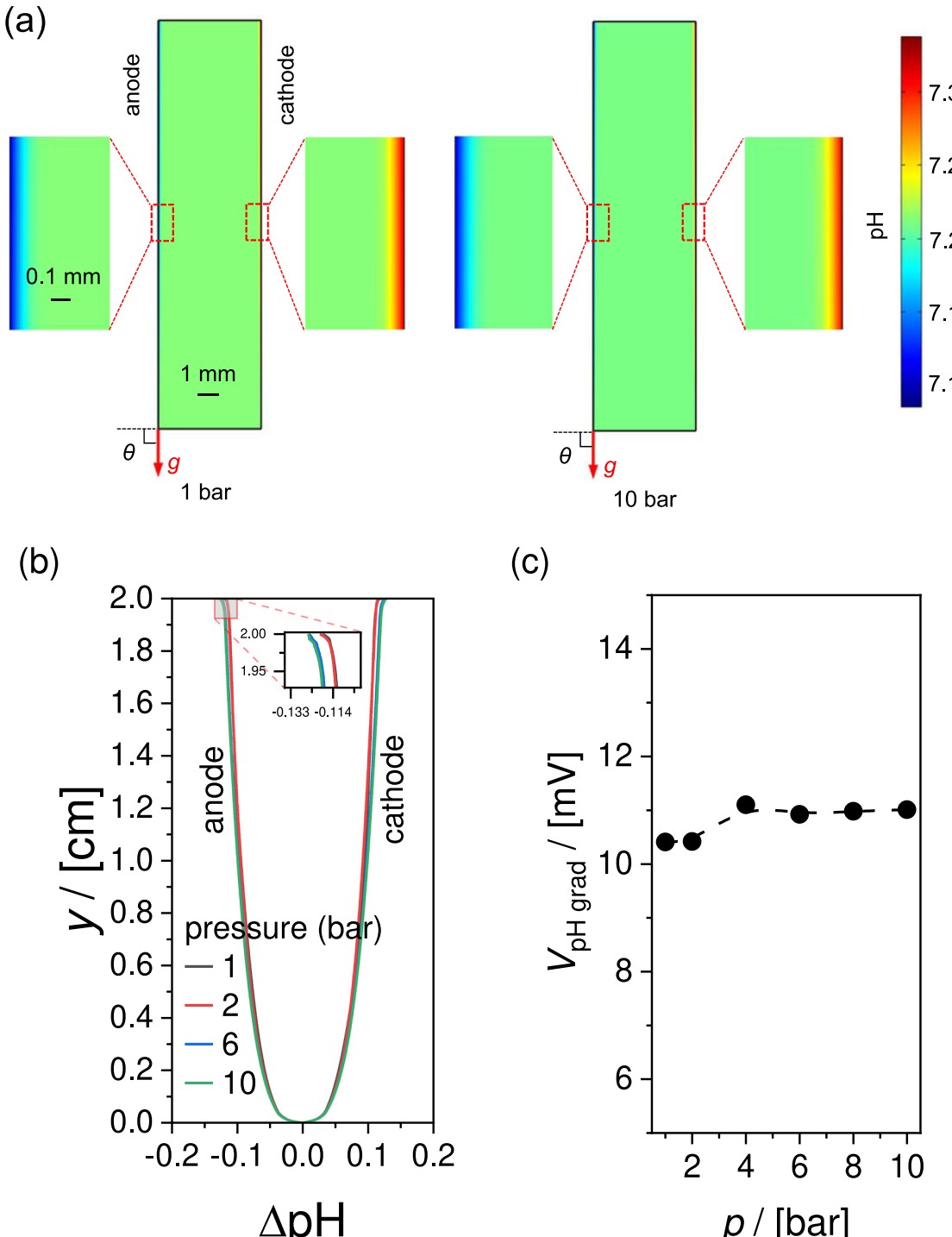

**Fig. 5 | Influence of operating pressure to the simulated local pH distribution and concentration overpotentials. a** Colormaps of pH in the cell at different operating pressures, i.e., 1 bar vs. 10 bar. **b** pH gradient near the electrodes at different operating pressures, **c** pH gradient induced voltage loss ($V_{\text{pH grad}}$) vs. pressure. The current density is 10 mA cm⁻², the inlet electrolyte velocity is 3 cm s⁻¹, the device tilt angle is 90°. The electrolyte is 2 M KP$_i$ (pH 7.2).

translation from the various units to kWh are provided in the "Methods" section.

The breakdown of various energy losses as a function of the operating pressure is shown in Fig. 6b. The cumulative losses at 1 bar (-5.5 kWh) represent the losses associated with operating under the first scenario of Fig. 6a. Here, the main losses are bubble-induced optical losses (-3 kWh) and the energy needed to compress H₂ to 700 bar (-2 kWh). With increasing operating pressure, the second scenario of Fig. 6a comes into play. Both the bubble-plume-induced

optical losses ($f_{\text{opl}}$) and product crossover losses ($f_{\text{col}}$) are suppressed as the pressure increases. The minor contribution of $f_{\text{col}}$ is interesting considering that our simulations assume a membrane-free cell configuration; the inclusion of membrane or separator in the reactor will likely suppress $f_{\text{col}}$ even further. At the same time, the specific work for post-generation compression is also reduced with increasing PEC operating pressure (see Fig. S12). In contrast, the losses due to the increase in thermodynamic cell voltage ($f_{\text{tcv}}$) increase with pressure, while the concentration overpotential losses ($f_{\text{cop}}$) remain relatively

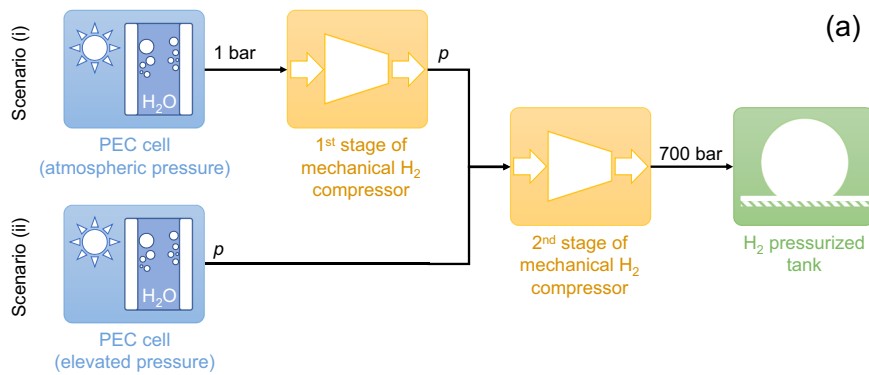

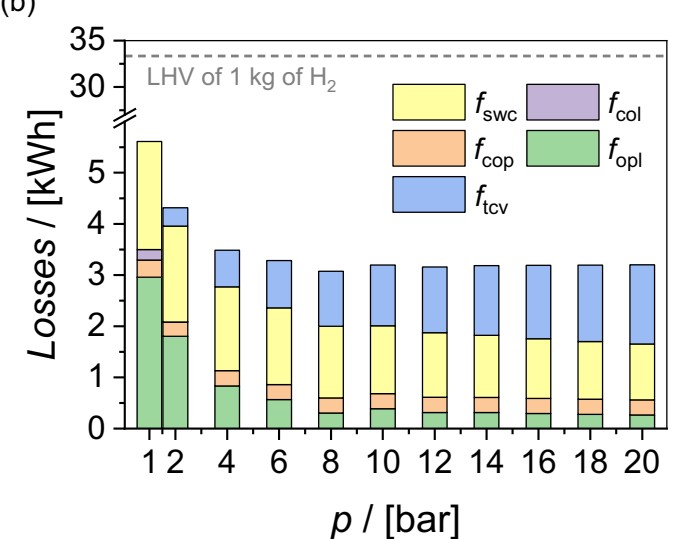

**Fig. 6 | Assessment of the benefits and penalties of operating PEC water splitting cells at elevated pressure. a** Schematic illustration of the two scenarios considered in our analysis. The two branches in the image represent different options for performing PEC water splitting, either at (i) atmospheric pressure or ii) at elevated pressure $p$. The collected hydrogen from both operation modes are fed into mechanical compressors in order to reach the pressure of 700 bar at the final storage tank. **b** Energy losses associated with operating PEC water splitting cells at various pressures. A 1 $m^2$ PEC water splitting device generating 1 kg of $H_2$ is considered. The gray horizontal dashed line indicates the lower heating value (LHV) of 1 kg of $H_2$ for a comparison. In panel (b), $f_{swc}$ is the specific work required for the compression of $H_2$ from collected to 700 bar, $f_{col}$ is the crossover loss induced by the gaseous products, $f_{cop}$ denotes the pH gradient induced voltage loss, $f_{opl}$ is the optical loss induced by bubble scattering effect, $f_{tcv}$ is the thermodynamic cell voltage loss associated with pressure elevation.

constant. The combined losses reach a value of ~3 kWh at an operating pressure of 6–8 bar, which represents a ~ 50% reduction compared to the losses at 1 bar. A further increase in pressure offers no significant advantages, as the total loss plateaus at the same level (± 5%). It is also interesting to note that even if the impact on optical loss is neglected, which may be realized in a certain device configuration (e.g., downward facing photoelectrode combined with upward facing dark electrode), the total energy loss at 6 bar is very close to that at 1 bar (see Fig. S13 in which Fig. 6b is re-plotted without considering $f_{opl}$). We therefore conclude that only minor benefits can be obtained by conducting PEC water splitting in a membrane-free cell beyond 6–8 bar, which should be considered as the optimum pressure range for the operation of such devices.

Finally, we note that our quantitative comparison is currently limited to the above-mentioned loss mechanisms and considerations. In reality, more factors need to be included in analyzing the feasibility of PEC water-splitting devices operated at elevated pressure, such as the mechanical strength of the cell materials, pumping power loss, the longevity of the (photo)electrodes and catalysts when operating at elevated pressure, the maintenance cost of the cell, etc. Our multiphysics model also assumes a constant cell temperature of 25 °C and

that the physical processes in the photoelectrodes are not affected by pressure elevation. In addition, although the pressure-induced changes of bubble characteristics are independent of the electrode-electrolyte system, the surface properties of the (photo)electrode (e.g., morphology, wettability, etc) and the concentration of the electrolyte could alter the bubble characteristics at a given pressure. These considerations, including bubble nucleation and growth, are not included in our Euler-Euler model. Including these contributing factors in the current analysis is not trivial, as it requires detailed knowledge of the device configuration and construction. As no elevated pressure PEC has yet been demonstrated, these details are currently not available. We have initiated efforts to directly measure the various loss contributions, i.e., optical loss, crossover, and pH distribution (using e.g., in situ fluorescence techniques we reported previously[69]) in our pressurized flow cell (Fig. S4); however, these measurements are not yet technically feasible due to the additional complexity introduced by elevated pressure operation. Further efforts to construct and study higher-pressure PEC devices, even at the lab scale, are therefore urgently needed in order to experimentally unravel pressure-dependent parameters and fully evaluate the potential of PEC water splitting at elevated pressure.

In summary, we have evaluated the quantitative benefits and penalties for operating a PEC water-splitting device at elevated pressure. This was accomplished by developing a multiphysics model that combines electrochemistry, two-phase fluid flow, and diluted species transport considerations in a membrane-free configuration. A key component of the model is the use of pressure-dependent bubble characteristics, which were obtained from empirical data that was validated against our experimental observations of $O_2$ bubbles in a moderately pressurized water-splitting cell. The operation at elevated pressure (up to 10 bar) was then simulated using our model, revealing the influence of operating pressure on the various loss factors in a PEC water-splitting device. Most importantly, bubble formation and the spatial extent of the bubble plume are minimized with increasing pressure. This, in turn, suppresses product crossover to a negligible level at pressures > 2 bar, while bubble-induced optical losses can be minimized to ~1% at 10 bar. The positive impact of bubble-induced convection to minimize local pH gradient was found to diminish with increasing pressure; this effect is, however, not significant even for a neutral pH electrolyte (the change in concentration overpotential was ~1 mV for a pressure elevation up to 10 bar). These insights were used to evaluate the combined energy losses in a scenario where the PEC-generated hydrogen was further compressed to 700 bar, taking into account the energy needed for this post-generation compression step. We showed that by increasing the operating pressure of the PEC cell from 1 to 6–8 bar, the combined energy losses can be reduced by a factor of ~2. Our analysis revealed that there is no measurable benefit from operating beyond 6–8 bar, which is therefore proposed as the optimum pressure range for the operation of PEC water-splitting devices.

## Methods

### Bubble observation in a pressurized water-splitting cell

A customized (photo)electrochemical cell printed using transparent heat-resistant plastic (VisiJet M2S-HT90) was used as the water-splitting cell, which has an internal volume of ~120 ml, with an electrode active area of 2 cm × 2 cm. This cell was operated safely up to a pressure of 4 bar(a). Note that this limited pressure range should not be confused with that used in our simulations, which is 1–10 bar (or 20 bar for certain cases). The technical drawing of this cell is available from the corresponding author upon reasonable request. The electrodes were prepared by depositing 5 nm of Pt onto FTO substrates (thickness of 2.2 mm, ~7 Ω/sq, Sigma-Aldrich). The deposition was done by electron beam evaporation in a customized high vacuum deposition chamber (Telemark) with a base pressure of $2 \times 10^{-7}$ mbar. A deposition rate of $0.5\,\text{Å}\,\text{s}^{-1}$ was used. The Pt/FTO electrodes were used as both the anode and cathode. To form an electrical connection with the electrodes, a small part of the Pt surface was connected to an electrical wire with a conductive tape (3 M). An aqueous solution of 1 M KOH (Merck KGaA, ≥85%) was prepared as the electrolyte using ultrapure water (resistivity > 18.2 MΩ cm, Milli-Q Integral system). The electrolyte was purged with $N_2$ for ~10 min while stirred using a magnetic stirrer before each measurement. The water-splitting cell was pressurized by feeding $N_2$ into the cell while controlling the mass flow rate at the cell outlet using a pressure controller (Bronkhorst High-Tech, uncertainty: 0.2%). Electrochemical measurements were performed in a two-electrode configuration using a VersaSTAT 3 potentiostat/galvanostat (AMETEK). The current density was kept constant at 10 mA cm⁻². $O_2$ bubbles were recorded using a shadowgraphy setup from LaVision. A solar simulator (Wacom-WXS-100S-L2H AM 1.5GMM) was used to illuminate the electrodes; the shadow of the bubbles could therefore be detected using a camera (2752 × 2200 pixels, 25 fps) placed behind the electrodes. A Makro-Planar T* 2/100 lens (Zeiss) was used to capture the bubble shadow graphs, and the aperture was kept as f = 2 (minimum value). This setting enabled us to focus our view on bubble formation from the chosen electrode. Bubble formation from

the other electrode did not interrupt our observation due to the relatively large distance between the two electrodes (~3 cm). See the details of the schematic illustration and the setup image in Fig. S1a, b. 500 image frames at 25 fps were recorded (i.e., acquisition time of 20 s) ~5 mins after applying the constant current density to ensure that steady state condition of bubble formation has been reached. The images were processed and analyzed using ImageJ. An illustration of the captured $O_2$ bubbles is shown in Fig. S1c. A region-of-interest (ROI) in the optical frame was selected to obtain the average bubble diameter, $D_{bub}$ and number density of bubbles, $N_{bub}$. Regions where bubbles are active, i.e., frequently formed, grew, detached, etc., were selected for our analysis. In practice, three factors were considered when choosing the ROI: (i) the size of bubbles within the region is uniformly distributed, (ii) the region does not contain a large accumulation of bubbles, and (iii) the region is located near the center of the frame. Due to the limitation of our cell design (no forced convection), electrode regions near the edges (especially the top edge) are more likely to contain large accumulation of bubbles, see Fig. S1c. Therefore, these regions were avoided when choosing the ROI. Following the method described in Supplementary Note 1, the experimental bubble formation efficiency, $\eta_{bub}$, was derived, except that the rising velocity of bubbles captured with the time-sequential frames from our measurements was used instead of the Stoke terminal velocity shown in equation S3.

### Multiphysics model

**Two-phase fluid flow.** An Euler-Euler laminar model was adopted to simulate the two-phase fluid dynamics in our membrane-free PEC water-splitting device. As depicted in Fig. S3a, b, the governing equations, including two sets of continuity and momentum equations, were solved in a 2D domain.

$$\frac{\partial(\alpha_i \rho_i)}{\partial t} + \nabla \cdot (\alpha_i \rho_i \mathbf{v}_i) = 0 \tag{2}$$

$$\alpha_i \rho_i \left( \frac{\partial \mathbf{v}_i}{\partial t} + \mathbf{v}_i \cdot \nabla \mathbf{v}_i \right) = -\alpha_i \nabla p + \alpha_i \mu_{mix} \nabla^2 \mathbf{v}_i + \alpha_i \rho_i \mathbf{g} + \mathbf{F}_{i-j} \tag{3}$$

where $\alpha_i$, $\rho_i$, and $\mathbf{v}_i$ represent the volume fraction, density, and velocity vector of phase i (L for liquid and G for gas), respectively. $\mathbf{g}$ denotes the gravity vector. $\mu_{mix}$ is the dynamic viscosity of the mixture.

$$\mu_{mix} = \mu_L (1 - \alpha_G)^{-0.25} \tag{4}$$

The gravity vector in our 2D domain was defined based on the device tilt angle, $\theta$, and the gravitational acceleration constant, $\mathbf{g}$.

$$\mathbf{g} = -\mathbf{g}(\sin\theta, \cos\theta) \tag{5}$$

The volume fractions of the liquid and gas phases need to satisfy the following equation.

$$\alpha_G + \alpha_L = 1 \tag{6}$$

$\mathbf{F}_{i-j}$ in Eq. 3 is a momentum exchange term between two phases, and is described below.

$$\mathbf{F}_{i-j} = -\mathbf{F}_{j-i} = \frac{3}{4D_{bub}} C_D \alpha_G \alpha_L \rho_L |\mathbf{v}_G - \mathbf{v}_L|(\mathbf{v}_G - \mathbf{v}_L) \tag{7}$$

$C_D$ and $D_{bub}$ represent the drag coefficient and the bubble diameter, respectively. The Schiller-Naumann model was used to determine $C_D$.

$$C_D = \begin{cases} \frac{24}{Re_b}\left(1 + 0.15Re_b^{0.687}\right), Re_b < 1000 \\ 0.44, Re_b \geq 1000 \end{cases} \tag{8}$$

in which $Re_b$ denotes the particle Reynolds number,

$$Re_b = \frac{D_{bub}\rho_L|\mathbf{v}_G - \mathbf{v}_L|}{\mu_{mix}} \qquad (9)$$

$D_{bub}$ is the pressure-dependent bubble diameter. $Re_b$ was found to be $\approx 3$ when $v_L = 3\,cm\,s^{-1}$, $\theta = 90°$, and $p = 1\,bar$, indicating laminar flow.

The boundary conditions for the liquid phase include the following: (i) a fully developed laminar flow was assumed at the electrolyte inlet, and (ii) both sides of the channel were considered as a no-slip wall for the liquid phase. For the gas phase, bubbles were assumed to be generated uniformly along the electrode surface. At the electrode surface (see the gas-producing electrode in Fig. S3b), the inlet velocity of gas bubbles (in Eq. 10) was determined using the local current density ($\mathbf{j}_{loc}$) and pressure-dependent bubble formation efficiency ($\eta_{bub}$). The mass flux of gas bubbles was calculated using Eq. 11. All of these vectors were assumed to be perpendicular to the electrode surface, i.e., the x-direction.

$$\mathbf{v}_{G,x} = \frac{\mathbf{j}_{loc}\eta_{bub}M}{n_eF\rho_G} \qquad (10)$$

$$\mathbf{R}_{G,x} = \frac{\mathbf{j}_{loc}\eta_{bub}M}{n_eF} \qquad (11)$$

Here, $F$, $M$, and $n_e$ are the Faraday constant, the molar mass of gas ($O_2$ or $H_2$), and the number of electrons involved in the reactions, respectively. In our study, due to gravity, the overall product crossover was limited by the bubbles generated at the upward-facing electrode. The slip wall condition was set for bubbles at both sides of the channel.

## Mass transport and electrochemistry

Theory of diluted species was adopted to solve the mass transport of the dissolved ions, as depicted in Fig. S3c. Mass transport in the solution is governed by the Nernst-Planck equation:

$$\frac{dc_m}{dt} = -\nabla \cdot \mathbf{N}_m = -\nabla\left(-D_m\nabla c_m - \frac{z_mF}{RT}D_m c_m\nabla\phi_L + c_m\mathbf{v}_L\right) \qquad (12)$$

where $c_m$, $\mathbf{N}_m$, $D_m$, and $z_m$ denote the concentration, the molar flux, the diffusion coefficient, and the charge of the dissolved species, m, respectively. It is worth noting that the diffusion coefficient, $D_m$, is dependent on multiple parameters, see equation S15. We therefore checked the influence of pressure on $D_m$, see Supplementary Note 5 and Fig. S14. $\phi_L$ is the electrolyte potential. $\mathbf{v}_L$ is the electrolyte velocity and was determined from the two-phase fluid flow model. Electroneutrality was assumed, and conservation of charge was fulfilled in the electrolyte. The electrolyte pH was chosen to be the $pK_{a2}$ of the phosphate buffer since the local pH gradient is efficiently minimized (see the buffer equilibrium in the domain, Fig. S3c). Other minority phosphate buffer species, i.e., $H_3PO_4$ and $PO_4^{3-}$, were not considered in our model. The molar flux at the electrode surface was determined from the local current density:

$$\mathbf{n} \cdot \mathbf{N}_m = \frac{-v_m\mathbf{j}_{loc}(1 - \eta_{bub})}{n_eF} \qquad (13)$$

where $v_m$ is the stoichiometry coefficient, i.e., $-4$, $2$, and $-1$ for $H^+$, $H_2$ and $O_2$, respectively ($n_e = 4$). $\eta_{bub}$ is the pressure-dependent bubble formation efficiency. The local current density $j_{loc}$ was determined from the Butler Volmer equation:

$$\mathbf{j}_{loc} = \mathbf{j}_0\left\{\exp\left(\frac{\alpha_a F\eta}{RT}\right) - \exp\left(\frac{-\alpha_c F\eta}{RT}\right)\right\} \qquad (14)$$

where $\mathbf{j}_0$, $\alpha_a$ and $\alpha_c$ are exchange current density and the anodic and cathodic transfer coefficients, respectively. The overpotential, $\eta$ was determined from the following equations, which contain the concentration overpotential due to pH gradient.

$$\eta_{anode} = \phi_s - \phi_L - 1.23 + \frac{RT}{2F}\ln\frac{c_{H^+,bulk}}{c_{H^+}} \qquad (15)$$

$$\eta_{cathode} = \phi_s - \phi_L - 0 + \frac{RT}{2F}\ln\frac{c_{H^+,bulk}}{c_{H^+}} \qquad (16)$$

Here, $\phi_s$ denotes the electrode potential and is assumed to be constant along the electrode. The cathode potential is set to be 0 V. We note that the presence of bubbles may affect the local electrolyte conductivity, such that a Bruggeman correction may be required[70]. However, according to our previous study[29], including the Bruggeman correction does not change the local current density distribution (<0.1% over a 10 cm electrode) due to the relatively low operating current density (10 – 20 mA cm⁻²) in solar-driven water splitting and the compact cell size used in our study. Note that the effects of bubble adhering to the electrodes and increasing the overpotential is not directly simulated due to the limitations of our Euler-Euler model. Estimations were, however, made based on the gas volume fraction obtained from our model and an empirical correlation reported in the literature, as described in Supplementary Note 4.

## Numerical treatment

The multiphysics model was solved in a coupled mode: the liquid velocity determined by the Euler-Euler multiphase model contributes to the mass transport of the dissolved species, and the resultant local current density, in turn, determines the gas bubble inlet along the electrode in the Euler-Euler multiphase model. Steady state simulations were performed with COMSOL Multiphysics® 5.6 on a high-performance work station (Intel(R) Xeon(R) CPU E-5-2650 v2, 32 physical cores, 192 GB RAM). The fluid variables, such as pressure $p$, liquid (continuous) phase velocity $v_L$, and gas (dispersed) phase velocity $v_G$ were solved using the PARDISO solver, while the volume fractions $\alpha_i$ were solved with the MUMPS solver. The variables involved in mass transport and electrochemistry were solved with the PARDISO solver. A relative tolerance of 0.005 was set as the convergence criterion. The operational conditions of the membrane-free PEC water-splitting cell used in the simulations are tabulated in Table S2, and all the parameters used are summarized in Tables S3 and S4.

## Model and mesh independence validation

The two-phase model of this study is validated against experimental measurements, see the Supplementary Note S2, Figs. S4–S6 and Supplementary Movie 1–4. In addition, as declared in the report[32], our multiphysics model is largely inspired by previous studies of Mat and co-workers[71–73], in which they validated the simulated gas volume fraction against the measured data using local conductivity experiments. Mesh independence of the present model was checked by considering the relative error of the liquid velocity as a function of the max. mesh size, see Fig. S15.

## Quantitative comparison

**Crossover loss to kWh.** Based on the assumptions we made in the context, the theoretical production rate of $H_2$ can be obtained

$$\dot{\mathbf{m}}_{H_2,theory} = \frac{\mathbf{j}_{loc}M_{H_2}}{n_eF} \qquad (17)$$

where $j_{loc}$ is the average local current density, $M_{H_2}$ is the molar mass of $H_2$, $n_e$ is the electrons consumed in hydrogen production reaction, $F$ is the Faraday constant. The amount of time needed (in [h]) for

producing 1 kg $H_2$ out of PEC water-splitting cell according to our assumption is therefore,

$$t_p = \frac{m_{H_2}}{A_e \times 3600 \times \dot{\mathbf{m}}_{H_2}} \qquad (18)$$

where $m_{H_2}$ is the expected mass of $H_2$, $A_e$ is the active area of the electrode. With the crossover ratio $f_{cr,loss}$ obtained from our multiphysics model, the loss of $H_2$ can be obtained:

$$m_{H_2,loss} = m_{H_2} \eta_{bub} f_{cr,loss}[\%] \qquad (19)$$

where $m_{H_2}$ is the expected production mass of $H_2$, $\eta_{bub}$ determines the amount of $H_2$ in the gaseous phase among the overall production. Note that bubble-induced product crossover was solely considered in this study. The lower heating value (LHV) of $H_2$ was then used to translate the crossover loss to kWh:

$$f_{col}[kWh] = LHV \times m_{H_2,loss} \qquad (20)$$

**Optical loss to kWh**

The optical loss (in [%]) obtained from our model can be translated to kWh using the following relation:

$$f_{opl}[kWh] = f_{opt,loss}[\%] \times LHV \times m_{H_2} \qquad (21)$$

where $f_{opt,loss}[\%]$ is the optical loss that is obtained in our multiphysics model.

**Specific work for $H_2$ compression**

The specific work needed for $H_2$ compression can be determined using the equations[74]

$$f_{swc} = \frac{|d\bar{w}_{H_2}|}{M_{H_2}} \qquad (22)$$

$$d\bar{w}_{H_2} = \int_1^2 p\,d\bar{v} = \bar{R}T\left(\ln\frac{\bar{v}_2}{\bar{v}_1} - 1.438\times10^{-5}\left[m^3mol^{-1}\right]\left(\frac{1}{\bar{v}_2}-\frac{1}{\bar{v}_1}\right)\right.$$
$$\left. - 3.438\times10^{-10}\left[m^6mol^{-2}\right]\left(\frac{1}{\bar{v}_2^2}-\frac{1}{\bar{v}_1^2}\right)\right) \qquad (23)$$

where $f_{swc}$ is the specific work needed for compressing 1 kg $H_2$ from state 1 to state 2, which is already in [kWh].

**Concentration overpotential to kWh**

The concentration overpotential, $V_{pH,grad}[V]$, can be translated into kWh with the following relation:

$$f_{cop}[kWh] = 1\times10^{-3} \times \mathbf{j}_{loc} \times V_{pH,grad}[V] \times t_p \times A_e \qquad (24)$$

**Thermodynamic cell voltage loss to kWh**

The pressure-induced thermodynamic cell voltage loss, $E_{cell}^0[V]$, can be translated into kWh with the following relation:

$$E_{cell}^0(p,T) = E^0(T) + \frac{R \cdot T}{z \cdot F}\ln\left(\frac{a(H_2)\cdot\sqrt{a(O_2)}}{a(H_2O)}\right) \qquad (25)$$

$$f_{tcv}[kWh] = 1\times10^{-3} \times \mathbf{j}_{loc} \times E_{cell}^0[V] \times t_p \times A_e \qquad (26)$$

## Data availability

All raw data generated during the study are available from the corresponding author upon reasonable request. The data shown in Figs. 1, 2, 4–6, and Supplemental Figs. S2, S5–S7, and S9–S15 are provided in the Source Data file. Source data are provided in this paper. Source data are provided in this paper.

## Code availability

The multiphysics model files used in this study are available from the corresponding author upon reasonable request.

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

## Acknowledgements

The Helmholtz Association of German Research Centers (HGF) and the Federal Ministry of Education and Research (BMBF), Germany, are gratefully acknowledged for supporting the development of solar-powered technologies for $H_2$ generation within the frame of the Innovation Pool project "Solar $H_2$: Highly Pure and Compressed" and the Helmholtz Research Program "Materials and Technologies for the Energy Transition" (MTET). Part of the work was also carried out with the support of the Helmholtz Energy Materials Foundry (HEMF), a large-scale distributed research infrastructure founded by the German Helmholtz Association. We also acknowledge Karsten Harbauer for his assistance in the preparation of Pt/FTO electrodes, Christian Höhn and Markus Bürger for the construction of the pressurized water splitting cell, Dr. Keisuke Obata for his initial input on the multiphysics model, and Dr. Babu Radhakrishnan for his assistance during the validation experiments. F. F. A. acknowledges support from CityU Start-up Grant (project 9610621).

## Author contributions

Conceptualization: F. L. and F.F.A. methodology: F.L. and F.F.A. investi-gation: F.L. writing–original draft: F.L. writing–review & editing: F.L., R.v.d.K., and F.F.A.; supervision, R.v.d.K., and F.F.A.; funding acquisition, R.v.d.K., and F.F.A.

## Funding

## Competing interests

The authors declare no competing interests.
