## [Peer Review File · Nature Communications]

REVIEWER COMMENTS

Reviewer #1 (Remarks to the Author):

This paper conducts a model-based evaluation of photoelectrochemical water splitting at elevated pressure conditions. It provides a thorough analysis of the benefits and penalties associated with high pressure operation, taking into account the effects on bubble characteristics, product gas crossover, bubble-induced optical losses, and concentration overpotential. Operating electrically driven water splitting at high pressures is of significant interest in both academic research and industrial applications. While photoelectrochemical water splitting is still at a relatively low TRL, the research is forward-looking by developing a fundamental understanding of how elevated pressure affects device performance, and can be useful to inform future design of photoelectrochemical devices. However, there are a few issues that need to be addressed/clarified.

1. One major issue related to gas-evolving electrodes is that bubbles adhering to the electrodes will block the reaction site and cause the shielding effect, resulting in a localized surge in overpotential (activation and ohmic losses). This can also contribute to increased heterogeneity in other distributions within the device, and therefore affect the overall performance of the device. It is not clear in the paper how the bubble coverage on the electrode surface and the shielding effect have been included in the model.
2. In the paper, the amounts of dissolved gases in the electrolyte are calculated from the bubble formation efficiency. It would be useful to validate the calculated values by comparing them with the solubility of gases in the electrolyte to see the deviations.
3. The model used in the paper is developed for a membrane-free cell. However, most photoelectrochemical reactors use membranes/separators, and these membrane-based reactors are the probable scalable options considering the extensive experience in scaling up membrane-based electrolyzers. Additionally, the bubble characteristics are determined using thin-film Pt electrodes, which can exhibit surface properties (e.g., morphology, wettability, etc) very different from those of commonly used nanostructured (photo)catalysts. The difference in surface properties can lead to a very different dynamics of bubble formation and growth near the electrodes. A question therefore arises: to what extent can the conclusions of this paper be generally applicable to (photo)electrochemical water splitting?
4. The selected electrode length (2 cm) is notably small in the context of industry applications. How scalable are the conclusions drawn in the paper?

Reviewer #3 (Remarks to the Author):

The authors investigated the implications of operating PEC water splitting directly at elevated pressure and evaluate the benefits and penalties associated with elevated pressure operation through the development of a multiphysics model. The topic discussed is interesting. However, I am concerned about

the setup the authors used for this demonstration. As I see, the electrodes used are not photoelectrodes. Most of the investigation and conclusions also applied to water electrolysis system. Another main issue is most of the experimental details are not sufficient. The comments are shown below.

1. The authors utilized Pt-decorated FTO electrodes with a constant current to simulate photoelectrochemical water splitting under elevated pressure. However, I am concerned about its suitability. 1) it is difficult to use this electrode to reflect the effect on the light absorption. The light scattering also depends on the wavelength. 2) the current varied greatly if applied a constant voltage. Utilizing a constant current mode, the voltage will be adapted according which may lead to a great change in voltage causing some irreversible damages to the photoelectrode. Maybe the authors can add one experiment with constant voltage to investigate how will the elevated pressure affect the current. The conclusions here can also apply to the water electrolysis, except the optical loss. Maybe can change the title to a large range of electrochemical or photoelectrochemical system.
2. In the method, the cell was operated safely up to a pressure of 4 bar against atmospheric pressure. However, the experiments are conducted under 1-10 bar. How could it operate above 4 bar?
3. According to Fig. S1b, the both O₂ and H₂ bubbles are generated at the same cell. How did the authors get the only O₂ bubbles visualization in Fig. S1c?
4. Fig. 1 compares the calculated values from the fitting equations with the experimental 131 dataset reported by Sillen (black data points). Why did the authors not include their own experimental data?
5. The authors define product crossover as the fraction of the molar flux that extends beyond the center of the channel. How did the authors measure the molar flux that extends beyond the center of the channel? These information should be described in the methods with detail.
6. How did the authors measure the optical loss in Fig. 7?
7. Again, how did the authors monitor the pH changes in Fig. 8b?

Response to reviewers' comments

Manuscript ID: NCOMMS-24-00792

Title: Model-based evaluation of photoelectrochemical water splitting at elevated pressure

In this response letter, the reviewers' comments to our original manuscript are provided in **black**, and our point-by-point responses as well as the corresponding changes to the manuscript are shown in **blue**.

Reviewer #1 (Remarks to the Author):

This paper conducts a model-based evaluation of photoelectrochemical water splitting at elevated pressure conditions. It provides a thorough analysis of the benefits and penalties associated with high pressure operation, taking into account the effects on bubble characteristics, product gas crossover, bubble-induced optical losses, and concentration overpotential. Operating electrically driven water splitting at high pressures is of significant interest in both academic research and industrial applications. While photoelectrochemical water splitting is still at a relatively low TRL, the research is forward-looking by developing a fundamental understanding of how elevated pressure affects device performance, and can be useful to inform future design of photoelectrochemical devices. However, there are a few issues that need to be addressed/clarified.

1. One major issue related to gas-evolving electrodes is that bubbles adhering to the electrodes will block the reaction site and cause the shielding effect, resulting in a localized surge in overpotential (activation and ohmic losses). This can also contribute to increased heterogeneity in other distributions within the device, and therefore affect the overall performance of the device. It is not clear in the paper how the bubble coverage on the electrode surface and the shielding effect have been included in the model.

Response: We thank the reviewer for this valuable suggestion. The effects of bubble adhering to the electrodes were not included in our model, due to the limitations of our Euler-Euler model in simulating the gas/electrolyte interface. We have emphasized this in the *Methods* section in the revised manuscript. However, we have now provided an estimate of how pressure would affect bubble coverage on electrodes' surface and the resulting overpotential. We obtained the gaseous volume fraction from our multiphysics simulations and integrated along the surface of the electrode in order to obtain the bubble coverage. The data is shown in Fig. R1a, with bubble coverage (θ) decreasing significantly from ~30% at 1 bar to < 5% above 4 bar. We then used the empirical correlations reported in the literature by Vogt¹ in order to estimate the bubble coverage overpotential (V_{θ}), as shown in Fig. R1b. The bubble coverage overpotential is ~20 mV at 1 bar and decreased to < 5 mV for operating pressure of > 4 bar. We have now added a brief discussion in the revised manuscript and as Supplementary Note 4 in the revised supplementary information.

Figure R1. (a) Bubble coverage (Θ) of electrode surface as a function of pressure as obtained from our multiphysics simulations (Fig. 4a-e) at device angle (θ) 90° . (b) The resulting bubble overpotential (V_{Θ}) calculated from the data of Θ in (a) and the empirical relationship reported by Vogt.¹

Associated changes to the manuscript:

- *Results and discussions (Manuscript)*, page 20, line 353 – 363: “Another important bubble-induced loss is the overpotential due to bubble coverage (or blockage) of the (photo)electrode area. However, this overpotential is not directly solved in our Euler-Euler multiphase model, as described in the Methods section. Instead, we have estimated the bubble coverage overpotential indirectly by (i) integrating the gas volume fraction along the surface of the electrode to calculate the bubble coverage (Θ) and (ii) using the empirical correlation reported by Vogt to evaluate the bubble coverage induced overpotential (V_{Θ}).¹ More details on the steps are described in Supplementary Note 4. Θ and V_{Θ} are plotted as a function of pressure in Figs. S9a and b, respectively. Increasing pressure significantly suppresses Θ and V_{Θ} ; at operating pressure of 4 bar and higher, the value of V_{Θ} is negligible (< 5 mV). Note that bubble coverage-induced overpotential is significantly smaller than the pH gradient-induced overpotential. Due to this very minor contribution, V_{Θ} is neglected in the following discussions.”
- *Method (Manuscript)*, page 32, line 599 – 602: “Note that the effects of bubble adhering to the electrodes and increasing the overpotential is not directly simulated due to the limitations of our Euler-Euler model. Estimations were, however, made based on the gas volume fraction obtained from our model and an empirical correlation reported in the literature, as described in Supplementary Note 4.”
- Supplementary Note 4, describing the steps to estimate bubble coverage and bubble coverage overpotential, has been added to the revised supplementary information.
- Figure R1 has been added as Figure S11 in the revised supplementary information.

2. In the paper, the amounts of dissolved gases in the electrolyte are calculated from the bubble formation efficiency. It would be useful to validate the calculated values by comparing them with the solubility of gases in the electrolyte to see the deviations.

Response: We appreciate the reviewer’s suggestion. Fig. R2 shows the comparison between the simulated gas concentrations from our model (black data points) with (i) the solubility of gases in the electrolyte (calculated using Henry’s law – blue data points) and (ii) the supersaturated gas concentrations as determined based on reported empirical correlations (red data points).^{2,3} It can be observed that the

concentration of dissolved gases increases with pressure, and our simulated results agree well with the empirical correlations. The slight discrepancy can be explained by the absence of gas bubble formation in the empirical correlation. We have added a brief discussion in the model validation section of Supplementary Note 2.

Figure R2. Supersaturated concentration of (a) O₂ and (b) H₂ gases in the vicinity of electrodes. c_w is the predicted concentration of O₂ or H₂ in the vicinity of electrode based on the empirical equation in literature.^{2,3} c_s is the molar solubility of gas in water, which is calculated based on Henry's law: $c_s = p_g/K_H$, in which p_g is the partial pressure of the products. Henry's constants for oxygen and hydrogen are $K_{H,O_2} = 769.2 \text{ atm M}^{-1}$ and $K_{H,H_2} = 1282.05 \text{ atm M}^{-1}$, respectively,^{4,5} assuming that the solution is pure water and the temperature is 298 K. $c_{O_2, \text{our model}}$ and $c_{H_2, \text{our model}}$ are the simulated O₂ and H₂ concentrations using our model, respectively. j is the current density, which is 10 mA cm^{-2} in this case.

Associated changes to the manuscript:

- *Supplementary Note 2 (Supplementary information):* “We also validated our model by comparing the dissolved gas concentration to that obtained from the empirical relationship reported by Shibata and Vogt.^{4,5} As shown in Fig. S6, both the dissolved O₂ and H₂ concentrations obtained from our model are much higher than the solubility values, and they are in relatively good agreement with the empirical supersaturated concentration; slight deviation can be explained by the fact that no bubble formation is considered in the empirical correlation.”
- Figure R2 has been added as Figure S6 in the revised supplementary information.

3. The model used in the paper is developed for a membrane-free cell. However, most photoelectrochemical reactors use membranes/separators, and these membrane-based reactors are the probable scalable options considering the extensive experience in scaling up membrane-based electrolyzers. Additionally, the bubble characteristics are determined using thin-film Pt electrodes, which can exhibit surface properties (e.g., morphology, wettability, etc) very different from those of commonly used nanostructured (photo)catalysts. The difference in surface properties can lead to a very different dynamics of bubble formation and growth near the electrodes. A question therefore arises: to what extent can the conclusions of this paper be generally applicable to (photo)electrochemical water splitting?

Response: We thank the reviewer for these valuable comments/suggestions. We are confident that our conclusion can be generally applied to (photo)electrochemical water splitting due to the following arguments. First, the introduction of membrane would only affect the product crossover loss (f_{col}), i.e., f_{col} would diminish in the presence of membrane. Since f_{col} has been shown to be a minor factor in the overall loss contributions (see Figs. 9b and S13), the inclusion of the membrane will not alter the pressure-dependent loss analysis. Second, regarding the influence of surface properties, we agree that bubble nucleation, growth and detachment will indeed vary depending on the electrode's surface roughness and/or wettability. These processes are not captured in our model and may be affected at higher pressure. However, we would like to emphasize that changes in bubble properties that we showed to be induced by pressure elevation will still likely be valid. This is supported by the good agreement between our measurements of bubble properties (performed using thin film Pt electrodes) and the empirical correlations obtained from Sillen's experimental data⁶ (performed using optically transparent and perforated plate nickel electrodes). Nevertheless, we acknowledge that our study and our multiphysics model have certain limitations. We have clearly described these limitations in the original manuscript, and we have now added additional limitations regarding bubble nucleation/growth in the revised manuscript.

Associated changes to the manuscript:

- *Results and discussions (Manuscript), page 24, line 420 – 424: “Both the bubble-plume-induced optical losses (f_{opv}) and product crossover losses (f_{col}) are suppressed as pressure increases. The minor contribution of f_{col} is interesting considering that our simulations assume a membrane-free cell configuration; the inclusion of membrane or separator in the reactor will likely suppress f_{col} even further.”*
- *Results and discussions (Manuscript), page 25, line 443 – 448: “In addition, although the pressure-induced changes of bubble characteristics are independent of the electrode-electrolyte system, the surface properties of the (photo)electrode (e.g., morphology, wettability, etc) and the concentration of the electrolyte could alter the bubble characteristics at a given pressure. These considerations, including bubble nucleation and growth, are not included in our Euler-Euler model.”*

4. The selected electrode length (2 cm) is notably small in the context of industry applications. How scalable are the conclusions drawn in the paper?

Response: We acknowledge that the electrode length considered in our study is small in the context of industrial applications. To assess the scalability of our results as suggested by the reviewer, we have now performed additional simulations using our multiphysics model considering larger cell geometries. L_x is fixed as 3 cm, while L_y is parameterized as 2, 5, and 10 cm, which is equivalent to an electrode active area of 4, 25, and 100 cm², respectively. We evaluated the influence of operating pressure on the simulated gas volume fraction. Indeed, as shown in Fig. R3, the gas volume fraction can be significantly suppressed at elevated pressure regardless of the cell geometry. This is consistent with the conclusion made from the smaller scale cell geometry, i.e., $L_x = 0.5$ cm and $L_y = 2$ cm, see Fig. 4 and Fig. 6 in the *Manuscript*.

Figure R3. Simulated gaseous phase distribution in various cell geometries at different pressure. (a) – (c) $L_y = 2, 4, 10$ cm at 1 bar. (d) – (f) $L_y = 2, 4, 10$ cm at 10 bar. L_x is kept as 3 cm. The current density is 10 mA cm^{-2} . The average inlet velocity of the electrolyte is set as 3 cm s^{-1} , and the device tilt angle is fixed at 30° . The scale bars represent 1 cm.

Reviewer #3 (Remarks to the Author):

The authors investigated the implications of operating PEC water splitting directly at elevated pressure and evaluate the benefits and penalties associated with elevated pressure operation through the development of a multiphysics model. The topic discussed is interesting. However, I am concerned about the setup the authors used for this demonstration. As I see, the electrodes used are not photoelectrodes. Most of the investigation and conclusions also applied to water electrolysis system. Another main issue is most of the experimental details are not sufficient. The comments are shown below.

1. The authors utilized Pt-decorated FTO electrodes with a constant current to simulate photoelectrochemical water splitting under elevated pressure. However, I am concerned about its suitability. 1) it is difficult to use this electrode to reflect the effect on the light absorption. The light scattering also depends on the wavelength. 2) the current varied greatly if applied a constant voltage. Utilizing a constant current mode, the voltage will be adapted according which may lead to a great change in voltage causing some irreversible damages to the photoelectrode. Maybe the authors can add one experiment with constant voltage to investigate how will the elevated pressure affect the current. The conclusions here can also apply to the water electrolysis, except the optical loss. Maybe can change the title to a large range of electrochemical or photoelectrochemical system.

Response: We appreciate the concerns raised by the reviewer and the valuable suggestions. We have now measured the transmission as a function of wavelength before and after bubble evolution at 1 bar. The result is shown in Fig. R4, indicating no wavelength dependence on the reduction on transmission. In other words, we can assume that bubble scattering effect is wavelength-independent, which is perhaps not surprising considering the relatively large bubble size compared to the optical wavelengths. In addition, we have also performed a constant voltage experiment as suggested by the reviewer and monitored the change in current with elevated pressure (Fig. R5). We observed a slight decrease of current, which is attributed to the increased thermodynamic cell potential at elevated pressure. Finally, we understand the reviewer's suggestion to change the title. Indeed, some of our conclusions are also applicable to electrochemical water splitting devices. However, considering that the bubble-induced optical loss is the dominating factor in the overall losses we considered (see f_{opl} in Fig. 9b), we believe the main messages of our study (e.g., optimum operating pressure) would still be more applicable to photoelectrochemical water splitting. Therefore, we have decided to keep the initial title of our manuscript.

Figure R4. Transmittance measured through a transparent (photo)electrochemical cell at 1 bar without and with bubble formation.

Figure R5. Current at various operating pressure. The working electrode is 5 nm Pt/FTO, the counter electrode is Pt wire, the reference electrode is Ag/AgCl. The electrolyte is 1 M KOH (pH = 14). Pressurization was done by purging O₂.

Associated changes to the manuscript:

- Supplementary Note 3 (Supplementary information): “Bubble scattering is assumed to be wavelength-independent, which is validated by measuring the transmittance through our (photo)electrochemical cell with and without bubble formation at 1 bar (see Fig. S9).”
- Figure R4 is added as Figure S9 in the revised supplementary information.

2. In the method, the cell was operated safely up to a pressure of 4 bar against atmospheric pressure. However, the experiments are conducted under 1-10 bar. How could it operate above 4 bar?

Response: We clarify that the experiments to validate our model were only performed up to 4 bar, as shown in Figs. 2, S1, S2, and S5. The results shown in our manuscript for 1 – 10 bar (or up to 20 bar for certain cases) were obtained by performing simulations with our validated model. To prevent any misunderstanding, we have now added a clarifying sentence to the *Methods* section.

Associated changes to the manuscript:

- *Methods (Manuscript)*, page 27, line 486 – 488: “This cell was operated safely up to a pressure of 4 bar(a). Note that this limited pressure range should not be confused with that used in our simulations, which is 1 – 10 bar (or 20 bar for certain cases).”

3. According to Fig. S1b, the both O₂ and H₂ bubbles are generated at the same cell. How did the authors get the only O₂ bubbles visualization in Fig. S1c?

Response: We utilized a Makro-Planar T* 2/100 (Zeiss) lens for our bubble shadowgraphy, and the aperture was kept as $f = 2$ (minimum). This equipment setting enabled us to focus our view on bubble formation from the chosen electrode, as shown by the schematic in Fig. S1b (*Supplementary information*). Bubble formation from the other electrode did not interrupt our observation due to the relatively large distance between the two electrodes (~3 cm). Detailed information regarding the camera setup has now been added to the revised manuscript.

Associated changes to the manuscript:

- *Methods (Manuscript)*, page 28, line 505 – 510: “A Makro-Planar T* 2/100 lens (Zeiss) was used to capture the bubble shadowgraphs, and the aperture was kept as $f = 2$ (minimum value). This

setting enabled us to focus our view on bubble formation from the chosen electrode. Bubble formation from the other electrode did not interrupt our observation due to the relatively large distance between the two electrodes (~3 cm)."

4. Fig. 1 compares the calculated values from the fitting equations with the experimental 131 dataset reported by Sillen (black data points). Why did the authors not include their own experimental data?

Response: Indeed, as the reviewer mentioned, the experimental data reported by Sillen⁶ was used to empirically correlate bubble characteristics (diameters, density) to the operating conditions (pressure, current density, flow rate) and generate fitting equations. Note that these empirical correlations satisfactorily cover the whole pressure range we considered in this study, i.e., 1 – 20 bar. Our experiments were then conducted at a lower pressure range (i.e., 1 – 4 bar) to independently validate the empirical correlations. Including our experimental data in the initial fitting steps should not be done as it would introduce a circular argument fallacy.

5. The authors define product crossover as the fraction of the molar flux that extends beyond the center of the channel. How did the authors measure the molar flux that extends beyond the center of the channel? These information should be described in the methods with detail.

Response: First of all, we clarify that the product crossover data that we presented were simulated results from our multiphysics model (as we clearly identified in the caption of Fig. 6). We obtained this data by introducing an imaginary line at the center of the channel and integrating the molar flux of bubbles to the left and right of this imaginary line. In order to prevent any potential confusion from readers, we have further identified that the data shown in Fig. 6 were simulated in the figure caption title.

Associated changes to the manuscript:

- Caption for Fig. 6 (*Manuscript*): "**Simulated** O₂ bubble molar flux at various operating pressures. Simulated O₂ bubble molar flux at the device outlet under various operating pressures. The average velocity at electrolyte inlet is set as 3 cm s⁻¹, current density is 10 mA cm⁻², device tilt angle is 30°."

6. How did the authors measure the optical loss in Fig. 7?

Response: As mentioned in the caption of Fig. 7 (*Simulated optical loss due to O₂ and H₂ bubbles at various operating pressures*), the optical loss data shown in Fig. 7 was not measured but simulated. In order to prevent any further misleading, we have further modified the figure caption.

Associated changes to the manuscript:

- Caption for Fig. 7 (*Manuscript*): "**Simulated** bubble-plume-induced optical loss as a function of the operating pressure. Simulated optical loss due to O₂ and H₂ bubbles at various operating pressures. The average velocity at electrolyte inlet is set as 3 cm s⁻¹, device tilt angle is 90°, current density is 10 mA cm⁻²."

7. Again, how did the authors monitor the pH changes in Fig. 8b?

Response: We apologize for the confusion. The local pH distribution and the concentration overpotential data shown in Fig. 8 are not measured results, but they are simulated results from our multiphysics model. We have now clearly identified this in the caption for Fig. 8.

Finally, we would like to emphasize that all the simulated results were obtained using a model that has been validated against experimental data. We are currently working on directly measuring all these data (optical

loss, crossover, pH distribution using in situ fluorescence techniques we reported previously)⁷ in a pressurized flow cell. The schematic and the digital photographs of the flow cell are shown in *Supplementary Note 2*. However, the required elevated pressure operation has been proven to complicate the measurements. Additional efforts beyond the scope of the current work are required. We hope to be able to report these results as soon as we overcome the challenges and are confident with the obtained data.

Associated changes to the manuscript:

- Caption for Fig. 8 (*Manuscript*): “**Influence of operating pressure to the simulated local pH distribution and concentration overpotentials.** (a) Colormaps of pH in the cell at different operating pressures, i.e., 1 bar vs. 10 bar. (b) pH gradient near the electrode at different operating pressures, (c) pH gradient induced voltage loss vs. pressure. The current density is 10 mA cm^{-2} and the inlet electrolyte velocity is 3 cm s^{-1} . The electrolyte is 2M KP_i (pH 7.2).”
- *Results and discussions* (*Manuscript*), page 26, line 451 – 457: “We have initiated efforts to directly measure the various loss contributions, i.e., optical loss, crossover, and pH distribution (using e.g., in situ fluorescence techniques we reported previously⁶⁹) in our pressurized flow cell (Fig. S4); however, these measurements are not yet technically feasible due to the additional complexity introduced by elevated pressure operation. Further efforts to construct and study higher-pressure PEC devices, even at the lab scale, are therefore urgently needed in order to experimentally unravel pressure-dependent parameters and fully evaluate the potential of PEC water splitting at elevated pressure.”

References

- 1 Balzer, R. & Vogt, H. Effect of electrolyte flow on the bubble coverage of vertical gas-evolving electrodes. *J. Electrochem. Soc.* **150**, E11 (2002).
- 2 Shibata, S. Supersaturation of oxygen in acidic solution in the vicinity of an oxygen-evolving platinum anode. *Electrochim. Acta* **23**, 619-623 (1978).
- 3 Vogt, H. On the supersaturation of gas in the concentration boundary layer of gas evolving electrodes. *Electrochim. Acta* **25**, 527-531 (1980).
- 4 Takahashi, T. *et al.* Global sea-air CO₂ flux based on climatological surface ocean pCO₂, and seasonal biological and temperature effects. *Deep-Sea Res. II: Top. Stud. Oceanogr.* **49**, 1601-1622 (2002).
- 5 Smith, F. L. & Harvey, A. H. Avoid common pitfalls when using Henry's law. *Chem. Eng. Prog.* **103**, 33-39 (2007).
- 6 Sillen, C., Barendrecht, E., Janssen, L. & Van Stralen, S. Gas Bubble Behaviour during Water Electrolysis. in *Hydrogen as an energy vector* 328-348 (Springer, 1980).
- 7 Obata, K., Van De Krol, R., Schwarze, M., Schomäcker, R. & Abdi, F. F. In situ observation of pH change during water splitting in neutral pH conditions: impact of natural convection driven by buoyancy effects. *Energy Environ. Sci.* **13**, 5104-5116 (2020).

REVIEWERS' COMMENTS

Reviewer #1 (Remarks to the Author):

My comments have been well addressed.

Reviewer #3 (Remarks to the Author):

no further comments.